# Effects of Low Intensity Pulsed Ultrasound Stimulation on the Temporal Dynamics of Irradiated Bone Tissue Healing: A Histomorphometric Study in Rabbits

**DOI:** 10.3390/ijms232012426

**Published:** 2022-10-17

**Authors:** Bob Biewer, Felix Kleine-Borgmann, Gaël P. Hammer, Eric H. Rompen, Michel Mittelbronn, Pascale Quatresooz

**Affiliations:** 1Department of Periodontology and Oral Surgery, Faculty of Medicine, 4000 Liège, Belgium; 2Department of Life Science and Medicine (DLSM), University of Luxembourg, 4365 Esch/Alzette, Luxembourg; 3Luxembourg Center of Neuropathology (LCNP), 3555 Dudelange, Luxembourg; 4Department of Cancer Research (DoCR), Luxembourg Institute of Health (LIH), 1445 Strassen, Luxembourg; 5National Center of Pathology (NCP), Laboratoire National de Santé (LNS), 3555 Dudelange, Luxembourg; 6Luxembourg Centre for Systems Biomedicine (LCSB), University of Luxembourg, 4367 Esch/Alzette, Luxembourg; 7Department of Human Histology and Dermatopathology, University Hospital of Liège University (CHU), 4000 Liège, Belgium

**Keywords:** low intensity pulsed ultrasound, osteogenesis, osteoradionecrosis, bone regeneration, tibia, rabbit

## Abstract

The present study evaluated the influence of Low-Intensity Pulsed Ultrasound (LIPUS) on the regeneration processes of non-critical-size bone defects in irradiated and non-irradiated rabbit tibias. Bone defects were surgically created on both tibiae of six rabbits. The control group had no additional treatment. In one intervention group, one tibia was irradiated with 15 Gy in a single dose. A second group was treated with LIPUS, and a third with a combination of both treatments. The control samples showed 83.10% ± 17.79% of bone repair after 9 weeks, while the irradiated bone had regenerated significantly less during the same period (66.42% ± 29.36%). The LIPUS treatment on irradiated bones performed a 79.21% ± 21.07% bone fill and could not significantly improve the response compared to the non-treated irradiated specimens. However, LIPUS treatment on non-irradiated bone showed bone formations beyond the size defect (115.91% ± 33.69%), which was a highly significant increase when compared to the control group or any irradiated group. The application of ultrasound to healthy bone produced highly significant and enhanced bone formations with 36.70% more regenerated bone when compared to the same application on irradiated bone. LIPUS vibration stimuli may be considered as a promising complementary treatment approach in non-irradiated bone regeneration procedures to shorten the treatment and enhance bone healing. In irradiated bones, the effect of ultrasound application is less clear, and further studies are needed to refine the dynamics of the present results.

## 1. Introduction

Radiotherapy damages cancer cells but can also harm healthy cells in the treated area. Rapidly dividing cells, such as epidermal basal cells or bone marrow-derived hematopoietic cells, are the most affected. The mandible is a frequent cause of concern, as it is inevitably exposed to radiation during the majority of head and neck cancer treatments, making osteoradionecrosis a serious complication of these treatments. The hallmarks of radiation damage to bone include endothelial cell necrosis leading to thrombosis and hyalinisation of blood vessels, fibrosis of the periosteum, osteoblast and osteoclast necrosis, and subsequent medullary fat infiltration and fibrosis [1]. Bone microvascularisation causes progressive capillary occlusion, resulting in a hypovascular, hypocellular, and hypoxic organ that is unable to maintain normal cellular turnover and synthesis. Repair thus becomes difficult due to the irradiated bone’s impaired ability to perform normal healing processes. Under these conditions, bone becomes more susceptible to injury, infection, necrosis, and fracture.

After illuminating a promising conservative management strategy for mandibular osteoradionecrosis with a success rate of 48% in 21 patients, Harris [2] suggested the use of low-intensity pulsed ultrasound (LIPUS) to stimulate bone revascularisation. In addition, LIPUS has been demonstrated to increase the osteogenic differentiation of mesenchymal stem cells [3,4,5], stimulate osteoblast proliferation and differentiation [6,7], and modulate osteoclastic activity [8].

It has further been shown that ultrasound application can improve tissue perfusion by promoting angiogenesis [9,10,11] and accelerate the healing process of stress fractures [12]. Other studies have demonstrated ultrasound’s positive effect on bone growth by stimulating fracture healing, as shown in animal models [13,14,15,16,17], or by accelerating the normal repair process, as revealed in double-blind randomised controlled clinical trials involving tibial and radial fractures in humans [18,19].

Nevertheless, LIPUS therapy in bone healing remains a controversial topic, as some systematic reviews of human trials have shown high levels of inconsistency in the efficacy of LIPUS, particularly when it is used as an adjunct in acute fracture and stress fracture healing [20,21,22,23].

Based on these considerations, this histomorphometric study aimed to investigate the influence of LIPUS on the healing and repair of irradiated cortical bone in rabbits.

## 2. Results

No complications were seen during the postoperative healing periods, and thus all rabbits (n = 6) were included in the statistical analysis. Bone growth takes place in all four test conditions (Figure 1a,b); the newly generated bone undergoes remodelling, which differs over time and based on external factors such as radiotherapy and ultrasound stimulation.

The quantitative histomorphometric evaluation showed that the control bone samples generated 83.10% ± 17.79% bone repair after 8 weeks, while the irradiated bone had significantly less bone regeneration, over the same period (66.42% ± 29.36%; *p* < 0.001). Treatment with LIPUS resulted in significantly improved bone repair in both non-irradiated and irradiated bones. In non-irradiated bones, LIPUS treatment resulted in 115.9% ± 33.7% total bone repair, or +32.8% (*p* < 0.0001). In irradiated bones treated with LIPUS, repair was somewhat lower, with a bone repair of 79.2% ± 21.1%, or +12.8%, though still significantly higher than in irradiated bones without LIPUS treatment (*p* = 0.007; Figure 2). 

There was no significant retention of OTC (the first colour to be administered after 1 week) in any of the groups (Figure 3). OTC concentrations were highest in the control lesions (1.80%) and lowest after irradiation (1.24%). The LIPUS-treated samples of irradiated and non-irradiated bone had OTC integrations of 1.69% and 1.92%, respectively, which were significantly different from the control samples. Similar observations were made in the existing native bone. 

XYL integration values between 3.77 and 7.98% were measured in all experimental conditions. However, clear separation of XYL was not achieved, and overlap with CAL and ALZ could not be completely excluded (Figure 4).

After 6 weeks, approximately 16–24% of CAL integration was measured in control, irradiated, and ultrasound specimens. After ultrasound treatment, between 16.84 and 23.93% of CAL integration were clear signs of active bone growth in the regenerated area of the irradiated bone defects. Well-structured lamellas and channels and the transition from woven to lamellar bone were observed. At this point in the healing process, the regeneration activities in the native non-irradiated tibial bones were relatively high (C group: 5.98% and US group: 5.23%) compared to those in the native irradiated tibial bones, which were 50% less (R group: 2.06% and R+US group: 3.74%).

The final label, ALZ, made up the majority of the dye in the regeneration area. As indicated in Table 1, approximately 20–25% of bone growth occurs during this late stage. 

New bone appositions were also observed under the periosteum, around the pin holding the surgical guide (Figure 5). In the subperiosteal and subendosteal areas, accumulations of OTC were primarily observed, which were particularly noticeable in the non-irradiated conditions.

## 3. Discussion

The regeneration of bone is a complex biological healing process that involves a cascade of numerous cell types and genes [17,24,25,26]. The mechanism by which low-intensity ultrasound can accelerate bone healing is not clear. Previous studies have shown that ultrasound enhances angiogenesis [27], calcium incorporation into the differentiating cells [28], and aggrecan gene expression in the early callus [24,27]. Based on the available evidence, the primary targets of ultrasound appear to be the differentiating cells in the hematoma and periosteum and the subsequent invasion of vessels and osteogenesis. The findings of this study confirm early bone formation and accelerated maturation in LIPUS-treated defects.

Radiation has combined effects on the cellularity and vascularisation of bone, which affect the matrix and endothelial cells. Vascular fibrosis reduces vascularisation, compromising the vitality of bone and marrow cells and rendering the area sensitive to infection and necrosis.

The capacity of bone to regenerate reduces proportionally with increasing doses of radiation up to the limit of 30 Gy [29]. Observations in the bone marrow of rabbit tibiae have revealed that a dose above 10 Gy induces osteoblast destruction and a dose above 20 Gy causes osteoclast apoptosis.

Arnold et al. [30] reported that a radiation dose of 13 Gy hinders the regeneration of rat femur bone defects of 1.2 mm in diameter by reducing the cellular population below a critical level, while doses greater than 15 Gy induce the destruction of the blood clot and inhibition of the regeneration process by affecting cellular migration to regions more distant to the bone marrow.

These findings are corroborated by the results of this study; 15 Gy of radiation applied to rabbit tibiae significantly reduced the bone regeneration potential by 16.68% compared to the regeneration in the control tibiae. Ultrasound application may improve irradiated bone’s capacity to recover to levels similar to healthy bone (C group: 83.10% ± 17.79% vs. R+US group: 79.21% ± 21.07%). 

The technique of polychrome fluorochrome labelling is used to evaluate in vivo bone formation and remodelling processes at different time intervals [31,32,33]. A limited number of time intervals is generally available, as only up to four fluorescent colours can be distinguished by conventional imaging systems (Figure 4). In particular, in the red, orange, and yellow spectra, clear discrimination between colours is not always possible, necessitating the employment of a longer time interval between marker applications [33]. Given these considerations, the administration of markers in our study was performed over three stages and followed by a 12–14 day drug-free period.

In all experimental groups, very little of the first label was visible. This may be due to either elimination from the ongoing remodelling processes or a lack of lesion calcification at the time of administration, preventing the OTC from binding to the calcium. The latter hypothesis, however, is contradictory to the fact that early ossification centres under the periosteum and endosteum maintained their OTC label over the 8-week labelling period (Figure 6). This confirms, as shown in Figure 5, that lifting the periosteum or endosteum by maintaining a secluded space sufficiently stimulates new bone formation even outside the skeletal boundaries [34]. This apparently reactive bone growth takes place exclusively at the beginning of the healing process. This area is not crossed by Haversian channels and is thus not subject to the remodelling processes, allowing the OTC label to outlast the 8-week study period.

In this context, one might question whether and to what extent the pressure of the overlying tissues could have influenced the quantitative evaluation of new tissue growth and how the results would have been impacted if the defects had been protected with a membrane according to the principles of guided bone regeneration.

We observed significantly higher label integration in non-irradiated native bones than in irradiated bones (C: 19.96%, US: 19.29% vs. R: 9.48%, R+US: 12.75%). Figure 7 illustrates an irradiated native tibia, which gives the impression of a suffering tissue with small amounts of the two early labels (R group: 0.88%/2.16% and R+US group: 0.91%/2.23%). The small amounts of OTC and XYL are due to not only replacement processes but also a lack of integration. The changes in bone-mineral content and the relative amounts of calcium and phosphorus in irradiated bone have been previously studied in rats [35]. The described reduction in calcium and phosphate may help to explain the poor integration of the first labels observed in this study. From the sixth week, the difference between irradiated and non-irradiated native bones was slightly less pronounced, and it appears that certain levels of recovery of the irradiated sites could explain the relatively high integration of ALZ at that time. The LIPUS treatment appears to have had an additional positive effect, as evidenced by the particularly high ALZ value in the R+US group. 

When compared to the control group (Intercept), all labels demonstrated lower values at all time intervals in the native irradiated bone. In addition, the number of Haversian channels was slightly decreased in irradiated tibiae compared to non-irradiated tibiae. These results were expected and consistent with other observations describing decreased local vascularisation as a result of obliterative endarteritis, periarteritis, and inflammation after exposure to ionising radiation [36]. The LIPUS treatment elicited a significantly higher fluorochrome integration with better marker binding to calcium, all signs of enhanced mineralisation (∑_R+US_ 12.75%), than the label integration observed during healing in the irradiated control group (∑_R_ 9.48%).

In regenerated bone, the integration or remnant of the two early labels, OTC and XYL, in the irradiated bone remained less than that seen in the non-irradiated bone. For both labels, LIPUS treatment after irradiation appeared to provide a faster and more favourable impulse for healing.

Comparison analysis provided particular evidence of the stimulating effect of LIPUS after irradiation. All four labels of the R+US group series were significantly increased compared to those of the irradiated and non-treated group series.

The same was true for the non-irradiated bone, although non-significant reductions in the OTC and XYL labels were recorded after LIPUS treatment. These small decreases could be due to increased remodelling processes after 5–6 weeks that replaced the early OTC- and XYL-loaded bone formations.

## 4. Materials and Methods

In this study, LIPUS was performed on irradiated bone defects in rabbit tibiae in order to evaluate its influence on bone tissue healing in vivo. 

### 4.1. Animals and Treatments

Six adult female New Zealand white rabbits (*Oryctolagus cuniculus* f. domesticus) weighing about 4.87 ± 0.12 kg were used in this study, providing a total of 12 tibiae and 3 technical replicates for each experimental condition and allowing us to ensure reproducibility while minimising animal use. Apart from the fact that female rabbits reach skeletal maturity a bit earlier [37], we have chosen female rabbits as it is known that male rabbits hit the ground with their back legs in stressful situations to a much higher degree than female rabbits, leading even to “spontaneous” back leg fractures after any kind of experimental surgery. Since the present experimental protocol induced a severe reduction of the back bones’ resistance, female rabbits were chosen in order to reduce the risk of additional fracture. The rabbits were individually housed in stainless steel cages in isolated rooms with independent ventilation for two weeks prior to the experimental procedure to ensure their health and stability. The room temperature and humidity were standardised. The animals were subjected to a 12 h light/dark cycle and were allowed to have access to food and water ad libitum. 

Bone defects were surgically created on all twelve tibiae. Two holes were created in the cortical bone using a drill of 3 mm in diameter, performed at low speed (approx. 800 rpm) and under continuous saline irrigation. A titanium pin was placed mesial to the defects in order to position the surgical guide used to standardise and relocate the hole defects after the healing processes. Tibias were preferred to femurs because of the significantly easier surgical access, leading to fewer post-operative complications. The animals were then randomly allocated into four groups: the C group, or the control group, in which only the surgical procedure was applied with no additional treatment; the R group, in which the tibia was irradiated with a single dose of 15 Gy; the US group, in which the tibia was treated with LIPUS; and the R+US group, in which the tibia was irradiated with 15 Gy and treated with LIPUS (Table 2).

The right tibia of each rabbit was irradiated 3 days prior to surgery with a single dose of 15 Gy under general anaesthesia. The animals were deeply anaesthetised using intramuscular injections of xylazine hydrochloride (5 mg/kg of Rompun 2%, Bayer, Brussels, Belgium) and ketamine hydrochloride (35 mg/kg of Imalgène 1000, Merial SAS, Brussels, Belgium). The irradiation was delivered by an X-ray apparatus (Stabilivolt Siemens, 190 kV, 18 mA, HVL: 0.5 mm Cu) at a dose rate of 0.65 Gy/min, a source-to-skin distance of approximately 30 cm, and an irradiation field size limited to 12 × 16 cm. 

On the third post-irradiation day, all rabbits underwent a surgical procedure on both tibiae under strict aseptic conditions with gentle atraumatic techniques. General anaesthesia was induced with Rompun and Imalgène, as described above. A face mask delivered 2 L/min of oxygen to the spontaneously breathing animals, and a dose of 0.5 vol% isoflurane was added via a regular vaporiser to maintain anaesthesia. A local injection of articaine hydrochloride was administered to both sites before beginning surgery.

The skin was first shaved and washed with a mixture of 70% ethanol and iodine. A deep longitudinal skin incision was made on the medial portion of the proximal tibial metaphysis. A titanium pin was placed into the cortical bone of the medial proximal tibial metaphysis in order to maintain the placement of the surgical guide while drilling two holes of 3 mm in diameter into the bone under continuous saline irrigation. This allowed the treatment procedure to be standardised and the experimental area to be correctly located again after the healing process.

The periosteum and attached muscles were closed with 5.0 resorbable sutures before closing the skin with 4-0 sutures (VICRYL, Johnson & Johnson, New Brunswick, NJ, USA). During the first 72 h after surgery, all animals received a subcutaneous injection of carprofen (Rimadyl, Zoetis, Belgium) dosed at 4 mg/kg/day for postoperative analgesia. The animals were returned to their cages without restrictions on movement or diet.

LIPUS stimulation in the US and R+US groups began 48 h after surgery under sedation (ketamine at 20 mg/kg intramuscularly and xylazine at 1.5 mg/kg intramuscularly). Ultrasound was applied daily for 20 min using an Exogen 2000 apparatus (Bioventus LLC, Durham, NC, USA) that delivered a 200 µs sine wave burst of 1.5 MHz at a frequency of 1.0 kHz. The skin was shaved once a week to ensure close transducer contact via a coupling gel and no-air proliferation of the ultrasound waves.

The following four fluorochrome dyes were injected at 2-week intervals in order to ensure sequential follow-up measurements of the tissue growth (Figure 8). 

Oxytetracycline (OTC) 92.6 mg/mL (Terramycine 100, Pfizer A. H.): 30 mg/kg injected three times subcutaneously during the first week after surgery.Xylenol orange (XYL) sodium salt (Acros Organics™, Geel, Belgium): 90 mg/kg injected three times subcutaneously during the third week after surgery.Calcein green (CAL; Acros Organics, Belgium): 20 mg/kg injected three times subcutaneously during the fifth week after surgery.Alizarin complexone (ALZ; Acros Organics, Belgium): 30 mg/kg injected three times subcutaneously during the seventh week after surgery.

All the rabbits were sacrificed after 8 weeks with an overdose of pentobarbital under general anaesthesia. The removed tibial tissues were immediately fixed in a buffered formaldehyde/ethanol solution (one part formaldehyde neutralised with CaCO_3_ [50 g/L] and two parts 80% ethanol).

### 4.2. Histological Analysis

The fixed biopsies were embedded in methyl methacrylate resin, and serial sections with a thickness of 80–100 µm were prepared with a sawing microtome (Leica SP 1600, Nussloch, Germany), then ground and polished to yield undecalcified sections with a final thickness of 20–40 µm. Some sections were stained for bright-field analysis with Stevenel’s blue and van Gieson’s picrofuchsin, where osteoid tissue appears green and mineralised bone appears red. The remaining sections were left unstained for confocal analysis.

Bright-field analysis of the sections stained with Stevenel’s blue/van Gieson’s picrofuchsin was performed using a Zeiss Axio Lab A1 microscope mounted with a phototube and an Axiocam 305 colour video camera that enabled live picture acquisition.

The unstained fluorescent sections were evaluated using a confocal laser microscope (Zeiss Laser Scanning Microscope LSM 800) that scanned samples sequentially prior to assembling pixel information into an image. The laser module provided four laser lines (405, 488, 561, and 640 nm) for fluorescent dye excitation (Table 3). All image acquisition and treatment processes were completed with the Zeiss ZEN blue edition microscope software. Image analysis was performed using the open-source processing software ImageJ (version 2.0.0). A mask based on a threshold allowing us to separate the signal and the noise for each colour channel of the confocal image was created in ImageJ. The area of staining was measured over the entire picture in an unbiased and automated way and statistically analysed. The resulting quantitative histomorphometric measurements of osteogenesis and the label integration into the newly generated tissue reflecting the dynamics of the processes were presented as means and standard deviations. 

### 4.3. Statistical Analysis

Normality was tested by Shapiro–Wilk tests. Since normality could not be assumed for all groups, Wilcoxon signed-rank tests were used for pairwise comparisons. All statistical analyses were performed using the Statistical Analysis System software (version 9.4; Cary, NY, USA), and the level of significance was set to 5%.

## 5. Conclusions

This investigation has confirmed that radiation significantly reduces the vitality of bone and its potential to recover, as indicated by the low levels of regeneration and weak marker values in native and regenerated areas. However, LIPUS application led to an increase in regenerated bone and demonstrated a positive effect on the dynamics of the healing process. It may be considered as a promising complementary treatment approach in non-irradiated bone re-generation procedures, shortening the treatment and enhancing bone healing. Considering the limitations of this study, the effect of ultrasound application on irradiated bones is less clear. Further investigations, using specimens available at additional time intervals, may be useful to refine the dynamics of the present results with a clearer understanding of the acting mechanisms of ultrasound on irradiated bone.

## Figures and Tables

**Figure 1 ijms-23-12426-f001:**
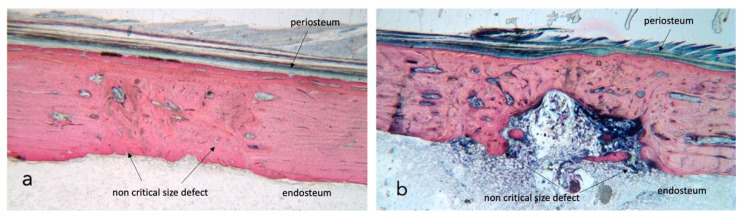
Stevenel’s blue/van Gieson’s picrofuchsin stained samples for light microscope analysis. (**a**) shows a control section with dense bone approaching 100% regeneration while in section (**b**) native tibia presents heavy damage by radiation and bone fill is incomplete. The external cortical is closed and covered by a continuous periosteum; the inner endosteal side presents a bone marrow ingrowth with signs of inflammation. ×2.5 magnification.

**Figure 2 ijms-23-12426-f002:**
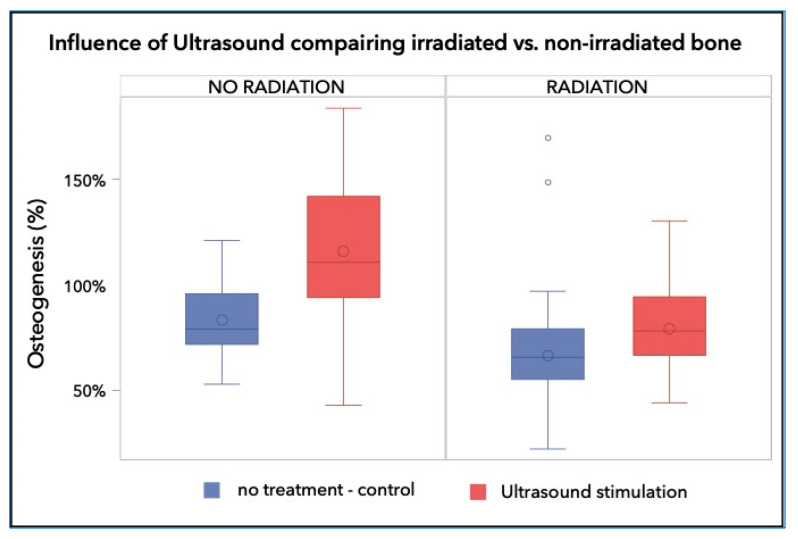
Percentage of the regenerated bone in the defect area of the control group was 83.10% ± 17.79%, whereas those of the irradiated groups reached a bone fill of 66.42% ± 29.36% and 79.21% ± 21.07%, respectively. The non-irradiated and LIPUS treated group achieved 115.91% ± 33.69%, which is a highly significant increase over the control and any irradiated group (*p* < 0.001).

**Figure 3 ijms-23-12426-f003:**
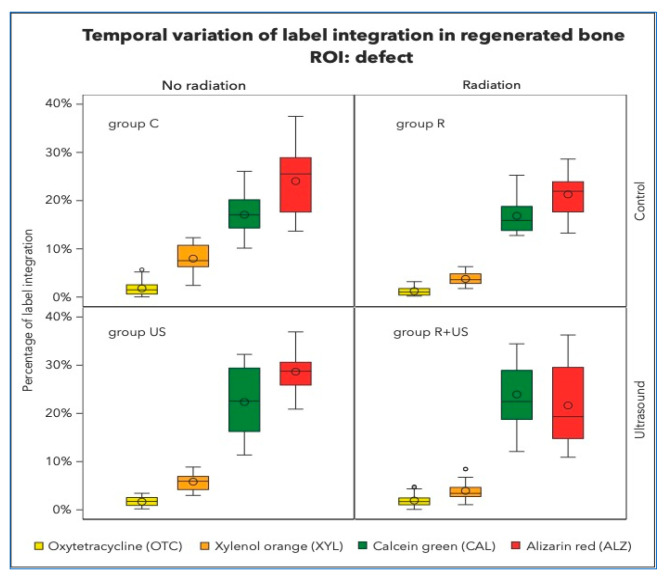
Percentage of label integration into the regenerated bone in the defects, i.e., region of interest (ROI) of each group. Note that the fluorescence spectra of Xylenol orange and Alizarin complexone are very close and hardly detectable to be clearly separated.

**Figure 4 ijms-23-12426-f004:**
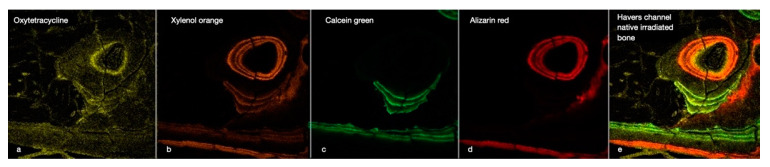
This image illustrates the chronology of the integration of the four fluorescent labels (OTC, XYL, CAL, and ALZ) into the cortical bone of an irradiated rabbit tibia (**a**–**d**). The image focuses on a Haversian canal containing a blood vessel surrounded by osteoblasts and osteocytes in their lacunae, with tubules connecting the cavities. Below are some periosteal bone attachments. The right window (**e**) gives an overall impression of bone activity over 8 weeks. (×40 magnification).

**Figure 5 ijms-23-12426-f005:**
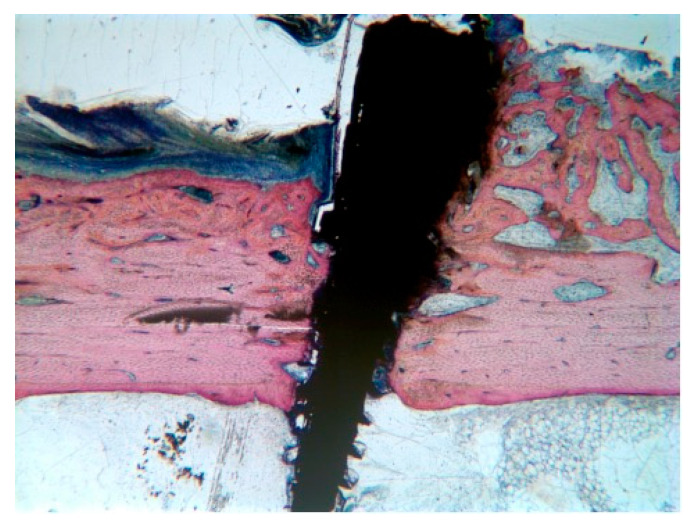
New bone appositions under the periosteum, around the pin holding the surgical guide. These bone formations were more likely found close to the surgically created defects. Here, the bleeding had formed a large blood clot supporting the periosteum for a longer time period, providing the necessary scaffold for new bone growth. Stevenel’s blue/van Gieson’s picrofuchsin stained sample at ×2.5 magnification.

**Figure 6 ijms-23-12426-f006:**
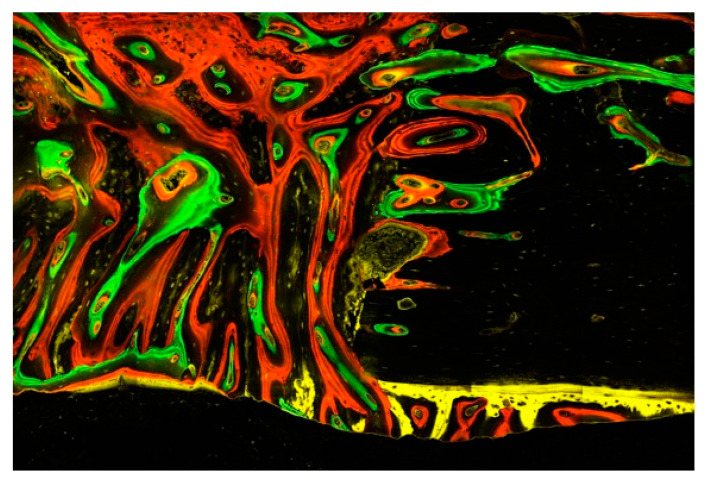
This image illustrates the different episodes of bone tissue formation documented by the four fluorescent dyes: the initial tissues are gradually replaced by connective tissue, including a multitude of mesenchymal cells and osteoblasts embedded in a fibrous matrix prior to mineralization into woven bone and finally lamellar bone with osteocytes. Of particular note are the remodelling processes emanating from the Haversian and Volkmann channels opened in the existing bone as well as the early mineralized tissue appositions underneath the periosteum, having been taken off during surgery. Evaluation with a confocal laser microscope (Zeiss Laser Scanning Microscope LSM 800) scanning fluorescent samples at ×40 magnification.

**Figure 7 ijms-23-12426-f007:**
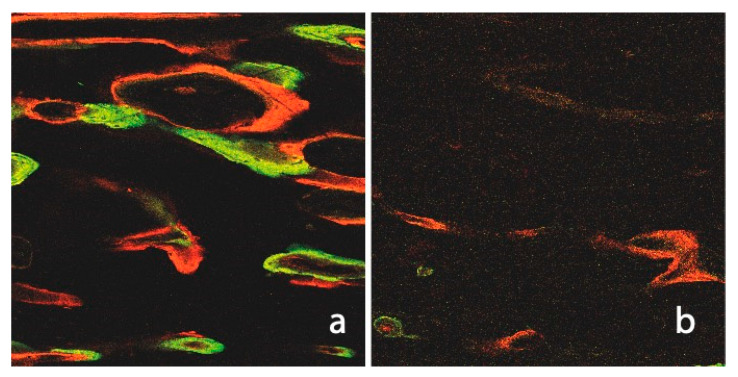
Fluorescence image of a non-irradiated and ultrasound of untreated native tibia (**a**). The numerous Haversian and Volkmann channels, as well as the intense label integration (CAL and ALZ), are notable when compared to the irradiated cortical bone (**b**).

**Figure 8 ijms-23-12426-f008:**
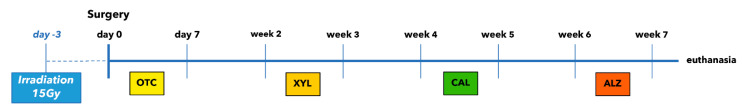
Time schedule of fluorescence labelling. Each marker has been applied three times.

**Table 1 ijms-23-12426-t001:** Percentage of fluorescence color labeling.

Comparison of the Effect of Ultrasound Percent Color Labeling		No Radiation					Radiation				
		Control (C)		Ultrasound (US)			Radiation Only (R)		Ultrasound (R+US)		
		Mean	SEM	Mean	SEM		Mean	SEM	Mean	SEM	
regenerated bone	OTC	1.80% ± 0.24%		1.69% ± 0.27%		**	1.24% ± 0.27%		1.92% ± 0.23%		**
	XYL	7.98% ± 0.45%		5.83% ± 0.50%		***	3.77% ± 0.39%		3.94% ± 0.34%		**
	CAL	17.07% ± 1.02%		22.35% ± 1.14%		***	16.84% ± 1.27%		23.93% ± 1.10%		***
	ALZ	24.04% ± 1.03%		28.65% ± 1.15%		***	21.29% ± 1.62%		21.64% ± 1.40%		***
native bone	OTC	1.93% ± 0.36%		1.97% ± 0.41%		**	0.88% ± 0.21%		0.91% ± 0.18%		*
	XYL	6.12% ± 0.61%		6.44% ± 0.68%		***	2.16% ± 0.31%		2.23% ± 0.27%		**
	CAL	5.98% ± 0.72%		5.23% ± 0.80%		**	2.06% ± 0.61%		3.74% ± 0.53%		**
	ALZ	5.93% ± 0.48%		5.65% ± 0.54%		***	4.38% ± 0.81%		5.87% ± 0.70%		**
		difference to control group C: statistically significant (*) *p* < 0.05—statistically highly significant as *p* < 0.001 (**) and *p* < 0.0001 (***).									

**Table 2 ijms-23-12426-t002:** Details on the four treatments randomly applied to the tibiae of six rabbits.

	Group C	Group US	Group R	Group R+US
Rabbit\Treatment	Only Surgical Procedure Was Applied Serving for Control	Low-Intensity Pulsed Ultrasound (LIPUS) Treated	15 Gy in Single Dose	15 Gy Irradiation and LIPUS Treated
1	Left tibia	-	Right tibia	-
2	Left tibia	-	-	Right tibia
3	-	Left tibia	-	Right tibia
4	-	Left tibia	Right tibia	-
5	Left tibia	-	-	Right tibia
6	-	Left tibia	Right tibia	-

**Table 3 ijms-23-12426-t003:** Overview of the four applied fluorochromes including administration, dosage, and spectral details with excitation and emission wavelengths and references. The confocal laser microscope settings for fluorescence signal capture are based on the author’s experience and previous works on polychrome labelling in bone tissue engineering research [38,39,40,41,42].

Fluorochrome	Dosage	Solvent	Admin	Excitation (nm)	Emission (nm)	Reference
Oxytetracycline	30 mg/kg	10% OTC sol	s.c	365–490	520–560	Lee T.C. 2003
Xylenol orange	90 mg/kg	NaHCO3 1%	s.c	440–570	610–615	O’Brien F.J. 2002
Calcein green	20 mg/kg	NaHCO3 2%	s.c	436–495	517–540	Pautke C. 2005, 2010
Alizarin Red	30 mg/kg	NaHCO3 2%	s.c	530–580	600–645	O’Brien F.J. 2002

## Data Availability

The data underlying this article will be shared on reasonable request to the corresponding author.

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
