# Peer review of "Effects of Low Intensity Pulsed Ultrasound Stimulation on the Temporal Dynamics of Irradiated Bone Tissue Healing: A Histomorphometric Study in Rabbits"

_ijms, 2022, doi:10.3390/ijms232012426_

Round 1

Reviewer 1 Report

This manuscript was well written and represents quite substantial research finding. According to this current study, radiation significantly lowers bone vitality, LIPUS application, on the other hand, increased bone regeneration and had a favourable impact on the dynamics of the healing process. Hence LIPUS, might be considered as a viable complementary therapy strategy for non-irradiated bone regeneration treatments that would speed up recovery and improve bone healing.

The arguments in the introduction and discussion sections comply with the manuscript objective. However, in order to improve the quality of the manuscript, listed below are my minor comments and suggestions:

1) The authors should give justification what is the main reason of choosing female rabbits instead of male rabbits. Is there any differences regarding the effects of radiation and LIPUS between female and male rats by referring at other articles?. Secondly, why is it tibia bone selected in this study instead of femur bone, since the femur is larger than tibia.

2) It is strongly advised to add the labelling for Figures 1, 4 and 5. Please enhance the quality of the histology images for Figure 4 as well as the text inside it.

3) In vivo should be in italic form (page 5, line 160 & page 7, line 232)

4) If it is applicable, in my opinion the authors should mention the normality test performed in their statistical analysis

Author Response

Response to Reviewer 1 Comments

First of all, we would like to thank the reviewers and editors for their comments and suggestions which have certainly been beneficial for our paper and for giving us the opportunity to submit a revised version of our manuscript.

Please find below our answers to the raised concerns:

Point 1: The authors should give justification what is the main reason of choosing female rabbits instead of male rabbits. Is there any differences regarding the effects of radiation and LIPUS between female and male rats by referring at other articles?. Secondly, why is it tibia bone selected in this study instead of femur bone, since the femur is larger than tibia.

Response 1:

Why females rather than males?

Apart from the fact that female rabbits reach skeletal maturity a bit earlier (Naff et al. 2012), we have chosen female rabbits as it is known that male rabbits hit the ground with their back legs in stressful situations to a much higher degree than female rabbits, leading even to "spontaneous" back legs fractures after any kind of experimental surgery. Since the present experimental protocol induced a severe reduction of the back bones' resistance, female rabbits were chosen in order to reduce the risk of additional fracture.

Why tibia instead of femur?

Tibias were chosen preferred to femurs because the surgical access is significantly easier at tibias, with much less muscles to incise and thus with much less post-operative complications.

Naff KA, Craig S. The Domestic Rabbit, Oryctolagus Cuniculus. In: The Laboratory Rabbit, Guinea Pig, Hamster, and Other Rodents. Elsevier 2012.

Point 2: It is strongly advised to add the labelling for Figures 1, 4 and 5. Please enhance the quality of the histology images for Figure 4 as well as the text inside it.

Response 2:

The suggested figures have been updated as follows.

Fig. 1: a more descriptive new labelling

Fig. 4: Enhanced illustration quality with a completely new labelling and legend

Fig. 5: Stevenel’s blue/ van Gieson's picrofuchsin stained sample at x2.5 magnification.

Point 3: In vivo should be in italic form (page 5, line 160 & page 7, line 232)

Response 3:

The recommended modifications have been done.

Point 4:  If it is applicable, in my opinion the authors should mention the normality test performed in their statistical analysis

Response 4:

According to the reviewer's request, we performed a normality test (Shapiro-Wilk). Since normality could not be assumed in our cohort, we have re-analyzed our data and replaced the previous test methods by the Wilcoxon test for pairwise comparisons. Of note, the p values only changed very marginally and stayed below 1%. We have adapted all p values that have changed in the revised version of our manuscript. We hope that this clarified all concerns of the reviewer regarding our statistical approach.

Reviewer 2 Report

The study by Biewer et al. investigates the effects of LIPUS on bone tissue healing using histomorphometry in rabbits. The manuscript reads well and beautiful figures of the fluorochromes are included. My strongest concern about the study is that its findings are very incremental to the already fairly abundant literature about LIPUS, fracture healing, and irradiation injuries. Below are some specific comments and suggestions.

Abstract:

·      Consider rephrasing the following sentence for clarity “..highly significantly higher..”.

Introduction:

·      Include the species in the aim e.g., “this histomorphometric study aimed to investigate 64 the influence of LIPUS on the healing and repair of irradiated cortical bone in [rabbits]”.

Materials and Methods:

·      Provide a mean and SD for the animals weight (not only an interval).

·      Justify the number of animals/tibiae by a sample size calculation and specify the power.

·      Specify what type of drill was used and how many RPM.

·      ® are redundant in scientific writing.

·      Table 3 (which is a Figure) is very illustrative for the study design.

·      There need to be more information about the histological assessment. Was assessment stereologically unbiased? Was a counting grid and or exclusion lines used?

·      The section about statistics needs to be expanded. Only variance is described to be assessed? I suppose the authors used an analysis of variance (ANOVA) to compare the three groups to Control (C)? If, so please provide details about post-hoc test, how normality, and homogeneity of variance was assessed?

Discussion:

·      “..Comparison by regression analysis provided…” the regression analysis is not mentioned in the section about Statistics.

General note:

·      The ARRIVE guidelines are not guiding principles of the practical conduct of studies using animals, but a reporting standard for what information to report in articles publishing data from in vivo studies. Moreover, the present manuscript does not adhere to the ARRIVE guidelines since not all items required to be reported are reported (e.g., sample size justification Item #2). Please carefully revisit the ARRIVE checklist to ensure everything in the manuscript is reported in accordance with ARRIVE.

Round 2

Reviewer 2 Report

My comments and suggestions have been addressed.